# Association of Pulmonary Function with Osteosarcopenic Obesity in Older Adults Aged over 50 Years

**DOI:** 10.3390/nu15132933

**Published:** 2023-06-28

**Authors:** Han-Sol Lim, Dong-Kun Kim, Hyun-Il Gil, Mi-Yeon Lee, Hyun-Seung Lee, Yong-Taek Lee, Kyung Jae Yoon, Chul-Hyun Park

**Affiliations:** 1Department of Physical and Rehabilitation Medicine, Kangbuk Samsung Hospital, Sungkyunkwan University School of Medicine, 29 Saemunan-ro, Jongno-gu, Seoul 03181, Republic of Korea; hslim0622@gmail.com (H.-S.L.); dongkun717@gmail.com (D.-K.K.); hyunseung429@gmail.com (H.-S.L.); yongtaek1.lee@gmail.com (Y.-T.L.); yoon.kjae@gmail.com (K.J.Y.); 2Division of Pulmonary and Critical Care Medicine, Department of Internal Medicine, Kangbuk Samsung Hospital, Sungkyunkwan University School of Medicine, 29 Saemunan-ro, Jongno-gu, Seoul 03181, Republic of Korea; hyunil.gil@samsung.com; 3Division of Biostatistics, Department of R&D Management, Kangbuk Samsung Hospital, Sungkyunkwan University School of Medicine, 29 Saemunan-ro, Jongno-gu, Seoul 03181, Republic of Korea; my7713.lee@samsung.com

**Keywords:** lung function, osteoporosis, sarcopenia, obesity, osteosarcopenic obesity

## Abstract

Osteosarcopenic obesity (OSO) is a newly described coexistence of osteopenia/osteoporosis, sarcopenia, and obesity. We examined the association between pulmonary function, OSO, and its composition in adults aged ≥ 50 years. A total of 26,343 participants (8640 men; 17,703 women) were classified into four groups based on the number of abnormal body compositions (osteopenia/osteoporosis, sarcopenia, and obesity): 0 (control), 1+, 2+, and 3+ (OSO) abnormal body compositions. The values of forced volume vital capacity (FVC)%, forced expiratory volume in 1 s (FEV1%), and FEV1/FVC% were significantly decreased by increasing the number of adverse body compositions (*p* < 0.0001). Although the prevalence of restrictive spirometry pattern (RSP) was positively associated with a higher number of abnormal body composition parameters (*p* < 0.001), obstructive spirometry pattern (OSP) had no association with adverse body composition. In multivariate analyses, the adjusted odds ratios (ORs) for RSP compared to the control group were 1.36 in 1+, 1.47 in 2+, and 1.64 in 3+ abnormal body compositions (*p* for trend < 0.001). Multiple abnormal body composition, especially osteosarcopenic obesity, was independently associated with poor lung function showing RSP in older adults over 50 years. The coexistence of these abnormal body compositions may be a predisposing factor for pulmonary function deterioration.

## 1. Introduction

Aging can cause many deteriorative changes, such as a reduction in bone tissue and muscle mass and a fat mass gain [1,2]. Through their functions as endocrine organs and the cellular linkages, bone, muscle, and adipose tissues are involved in the whole-body connections and consequently affect each other [3]. Osteosarcopenic obesity (OSO), a newly described term, is characterized by a concurrent deterioration of bone (osteopenia or osteoporosis), muscle (sarcopenia), and increase in fat (overweight or obesity) [4]. The development of OSO may be associated with low-grade chronic inflammation [3]. Recent studies have suggested that OSO is associated with decreased functionality, physical inactivity, low vitamin D, hypertension, and dyslipidemia [5,6,7].

There are restrictive and obstructive patterns in the spirometry pattern [8]. Restrictive spirometry pattern (RSP) is caused by the deficit of chest wall compliance, resulting in a reduction in forced expiratory volume in 1 s (FEV1) and forced vital capacity (FVC) but preserving the ratio of FEV1/FVC. In contrast, airway obstruction is commonly found in the obstructive spirometry pattern (OSP), and the FEV1/FVC ratio is significantly decreased [9]. Previous studies have presented that pulmonary function value is a strong indicator of mortality in the general public [10] and in cardiovascular disease [11]. Thus, it is necessary to identify which factors affect these values.

Another condition associated with aging is the worsening of lung function. Pulmonary function gradually declines after the age of 30 and decreases marginally more after 50 years [12]. In elderly individuals, the elastic components of the lung deteriorate, the parenchymal tissue is reduced, the lung cavity compliance decreases, the intercostal muscles are reduced, and the gas exchange surface lessens, resulting in poor respiration function [13]. Previous studies have reported that obesity is an important risk factor for many respiratory diseases [14,15,16]. Recently, low muscle mass was reported to be negatively associated with reduced lung function [17,18] and osteoporosis-related vertebral fracture and kyphosis can affect worsening lung function [19,20,21]. However, there have been no studies documenting the association of pulmonary function with OSO or combined abnormal body compositions.

The purpose of this study was to evaluate the relationship of combined abnormal body composition (bone, muscle, and fat) with values of spirometry and the prevalence of RSP and OSP in community-dwelling adults aged over 50 years.

## 2. Materials and Methods

### 2.1. Study Participants

This cross-sectional study investigated subjects in the Kangbuk Samsung Health Study from 1 January 2012 to 31 December 2018. Participants aged > 18 years performed a thorough medical screening at Kangbuk Samsung Hospital Healthcare Centers in Seoul and Suwon, Republic of Korea. Most of the subjects of KSHS are employees of numerous industrial companies because the Republic of Korea Industrial Safety and Health Law provides free health check-ups to employees every year or every other year. The residual subjects spontaneously conducted the health check-up programs. The purpose of health screening is to advance the health of employees through regular medical check-ups and to improve early diagnosis of existing diseases.

Individuals aged over 50 years who underwent dual-energy X-ray absorptiometry (DEXA), bioelectrical impedance analysis (BIA), and spirometry were included (*n* = 28,623) (Figure 1). The participants who have (1) a history of COPD (*n* = 449), (2) a history of asthma (*n* = 542), (3) a history of tuberculosis (*n* = 1343), and (4) a history of lung cancer (*n* = 39) were excluded. Accordingly, the remaining 26,343 participants were included in this cohort study (Figure 1). This study was conducted in accordance with the Declaration of Helsinki. Ethics approvals were obtained from the Institutional Review Board (IRB) of Kangbuk Samsung Hospital (Number: 2021-12-048). Informed consent was waived because researchers retrospectively achieved a de-identified database for analysis.

### 2.2. Determination of Adverse Body Composition and Categorization

Dual-energy X-ray absorptiometry (Lunar Prodigy; GE, Madison, WI, USA) was used to estimate the bone mineral density (BMD) of the lumbar spine (L2–L4) and total hip. Calibration and quality control were performed daily by trained medical technicians. Osteoporosis and osteopenia were defined as a T score of <−2.5 and −1.0 to −2.5, respectively, using the criteria of The World Health Organization (WHO) [22].

Appendicular skeletal muscle mass (ASM) measures and fat mass (%) were obtained with a BIA machine (InBody 720; Biospace, Seoul, Republic of Korea). The measurement was performed standing with both arms open from the body and holding the electrodes. The BIA was measured after overnight fasting to make the hydration condition as similar as possible and was calibrated from the beginning of all test days. Appendicular skeletal muscle mass was calculated as the sum of the muscle masses of all four limbs.

The skeletal muscle mass index (SMI) was calculated by dividing ASM (kg) by the height squared in meters (m^2^). Low muscle mass was defined as SMI < 7.0 in men and SMI < 5.7 in women, based on the Asian Working Group for Sarcopenia (AWGS) [23]. Obesity was defined as the percent of body fat mass > 35% in women and >25% in men according to a previous study [24].

All subjects were classified into the four groups based on the number of abnormal body compositions as normal (without osteopenia, sarcopenia, or obesity), one component (having one of the conditions), two components (having two conditions), and three components (OSO, having all the three conditions) based on previous studies [6,25].

### 2.3. Measurement of Spirometry and Definition of RSP and OSP

A spirometry test (without bronchodilator) was conducted by a Vmax instrument (Sensor-Medics, Yorba Linda, CA, USA). The FEV1 and FVC were obtained, and the highest values were used for the study. FVC is the volume of maximally exhaled air with maximum effort from a fully inhaled state. FEV1 is the maximal volume of air in the first second of an effortful expiration from a maximal inspiration state. The equations using a representative Korean population sample [26,27] were used to obtain the predicted values of FVC and FEV1. The percentage of predicted FVC and FEV1 were calculated by the following formula: FVC (L)/predicted FVC and FEV1 (L)/predicted FEV1.

The restrictive spirometry pattern (RSP) was defined as participants with a predicted FVC less than 80% and normal or increased FEV1/FVC [9]. Obstructive spirometry pattern (OSP) was defined as participants with FEV1/FVC < 0.7 and a predicted FEV1 < 80%, and no clinical symptoms [28].

### 2.4. Measurements of Biochemical and Clinical Parameters

The study data on age, sex, current smoker, alcohol use, exercise, and medical history (presence of hypertension, diabetes mellitus, and dyslipidemia) were collected using a self-administered questionnaire.

The questionnaire asked about the frequency and amount of alcohol consumption, smoking status, and exercise frequency. The questions about alcohol consumption included the frequency per week and the typical amount per day. Average amount of alcohol intake was calculated. Heavy drinking was defined as an average alcohol intake > 20 g/day. History of smoking was classified into never, ex-, or current smoker. Physical activity by the International Physical Activity Questionnaire Short Form (IPAQ-SF) was grouped into three categories: low, moderate, and high, and health-enhancing physically active (HEPA) was matched using IPAQ category II [29].

Presence of diabetes mellitus was defined as a fasting serum glucose ≥ 126 mg/dL, serum hemoglobin A1c (Hb A1c) ≥ 6.5%, a self-reported physician diagnosis, or current use of insulin or any antidiabetic medications using the diagnostic criteria of the American Diabetes Association and answers to the questionnaire [30]. Systolic blood pressure (SBP) and diastolic blood pressure (DBP) were measured using a standardized sphygmomanometer after resting for 5 min according to the Hypertension Detection and Follow-up Program protocol [31]. Hypertension was defined as having a measured blood pressure ≥ 140/90 mmHg, a self-reported physician diagnosis, or the current use of antihypertensive medication based on the criteria from the 8th report of the Joint National Committee [32,33].

Anthropometric measurements and laboratory tests were performed by trained nurses. Height and weight were measured twice with an electronic scale and averaged them. Body mass index (BMI) was defined as weight in kilograms divided by square of the height in meters (kg/m^2^). Waist circumference (cm) was measured at the midlevel between the lower border of the last palpable rib and the anterior iliac crest.

Blood samples were collected after a 12 h overnight fast. Serum total cholesterol, high-density lipoprotein cholesterol (HDL-C), low-density lipoprotein cholesterol (LDL-C), and triglycerides were measured with an enzymatic colorimetric assay. Serum fasting glucose level was measured using the hexokinase method on a Cobas Integra 800 system (Roche Diagnostics, Basel, Switzerland). Likewise, the hemoglobin A1c (HbA1c) levels were measured using an immunoturbidimetric method on the Cobas Integra 800 automatic analyzer (Roche Diagnostics). Serum insulin concentrations were measured with an electrochemiluminescence immunoassay (Roche Diagnostics).

All laboratory tests were conducted at The Laboratory Medicine Department of Kangbuk Samsung Hospital which has been accredited and participates annually in inspections and surveys by the Korean Association of Quality Assurance for Clinical Laboratories.

### 2.5. Statistical Analysis

The baseline characteristics of the study participants were presented as mean and standard deviation (SD) or percentages. The relationship between the characteristics and the number of adverse body compositions were compared using one-way analysis of variance (ANOVA) for continuous variables or chi-square tests for categorical variables.

ANOVA test was used to compare the mean values of spirometric parameters according to each group divided by the number of adverse body compositions. After adjustments for confounding variables such as age, sex, center, SBP, insulin, HDL-C, current smoker, heavy drinking, and physical activity (e.g., proportion of HEPA), analysis of covariance (ANCOVA) was used to assess differences of adjusted mean values [standard error (SE)] of pulmonary function parameters between adverse body composition groups with *p* value < 0.05. Post-hoc analysis was performed using the Bonferroni correction to compare the mean pulmonary function values between study groups. The prevalence of lung disease in each group classified according to adverse body composition was compared using the chi-square test.

Using binary logistic regression, the risk of lung disease according to the number of adverse body composition factors was evaluated as adjusted odds ratios (OR) and 95% confidence intervals (CI) after adjustments for confounding variables such as age, sex, center, SBP, insulin, HDL-C, current smoker, heavy drinking, and HEPA. A statistical significance was regarded at *p* value < 0.05. All statistical analyses were performed using SPSS Statistics version 26.0 (IBM Co., Armonk, NY, USA).

## 3. Results

### 3.1. Baseline Characteristics

The participants (*n* = 26,343) were divided according to the number of abnormal compositions as follows: (1) normal (*n* = 8497, 32.2%), (2) +1 (*n* = 11,895, 45.1%), (3) +2 (*n* = 5290, 20.1%), and (4) +3 (*n* = 661, 2.5%) (Table 1). The mean age of all subjects was 58.96 (SD, 6.71), and the prevalence of male sex was 32.8%. There were significant differences among the groups in all demographic and biomechanical variables (*p* < 0.05), except for prevalence of diabetes.

### 3.2. Pulmonary Function Values by Abnormal Body Compositions

Table 2 presents the spirometry values (FVC, predicted FVC%, FEV1, predicted FEV1%, and FEV1/FVC%) based on the number of abnormal body composition parameters. FVC, predicted FVC%, FEV1, and predicted FEV1% were negatively associated with an increasing number of abnormal body compositions (*p* < 0.001). These values decreased as the no. of abnormal body compositions increased (*p* < 0.001). After adjustments for possible confounding variables, including age, sex, center, SBP, insulin, HDL-C, current smoker, heavy drinking, and HEPA, significant associations of numbers of abnormal body compositions with FVC, predicted FVC%, FEV1, and predicted FEV1% remained persistent (Appendix A). The spirometry values were higher in the 0 (normal) groups than other abnormal body composition (1+, 2+, and 3+) groups, and the OSO (3+) groups had the lowest spirometry values among the four groups (*p* < 0.001). However, there was no significant linear association between FEV1/FVC% and abnormal body composition (*p* for trend = 0.756).

### 3.3. Prevalence of RSP and OSP by Abnormal Body Compositions

The prevalence of RSP was positively associated with an increasing number of abnormal body compositions: 22.9% in the 3+ group, 21.4% in the 2+ group, 19.5% in the 1+ group, and 13.2% in the control group (Figure 2; *p* for trend < 0.001). There was no significant linear trend in the prevalence of OSP with increasing abnormal body composition (Figure 2; *p* for trend = 0.060).

### 3.4. Association of RSP and OSP with Abnormal Body Compositions

Binary logistic regression analysis was performed to determine the ORs for the prevalence of RSP and OSP with respect to the number of abnormal body composition parameters (Table 3). In crude model, the risk of RSP compared with the control group was 1.59 (95% CI: 1.47–1.72) for 1+, 1.79 (95% CI: 1.63–1.96) for 2+, and 1.96 (95% CI: 1.64–2.33) for 3+ (*p* for trend < 0.001). After adjustments for possible confounding factors, compared to the control group, the adjusted OR (95% CI) of RSP were 1.36 (1.23–1.50) in 1+, 1.47 (1.30–1.67) in 2+, and 1.64 (1.27–2.11) in 3+ (*p* for trend < 0.001). However, the ORs (95% CIs) for OSP in one or more abnormal composition groups were not significant.

## 4. Discussion

This study of adults aged > 50 years without lung disease showed that multiple abnormal body compositions, especially OSO, are independently associated with poor long-term function. Furthermore, abnormal body composition was highly associated with RSP but not with OSP. This association remained significant after controlling for possible confounding factors, except for the FEV1/FVC value. To the best of our knowledge, this is the first study to report the association between OSO and pulmonary function in adults.

Several studies have compared the association between lung function and abnormal body composition. Ohara et al. reported that Brazil individuals over 60 years with sarcopenia had significantly lower FVC, FEV1, FEF 25–75%, and PEF than non-sarcopenic individuals [34]. According to earlier studies, when the respiratory system is impaired, physical activity decreases in elderly people, thereby increasing the risk of sarcopenia [17,18,35]. Previous reports showed that obese individuals had reduced lung volumes and capacities compared with normal weight individuals, which was speculated to be the result of the adipose tissue pressing the chest wall [15,36]. Hickson et al. reported that pericardial fat was associated with poor pulmonary function, showing RSP on spirometry [37]. Although the association between bone mass and pulmonary function is not well known, Harrison et al. reported that osteoporosis was directly related to vital capacity (VC) through secondary vertebral fracture or kyphosis [19]. It is a well-known fact that with aging there are changes in lung function, such as decreases in VC, FVC, and FEV1 [13]. OSO is characterized by a simultaneous deterioration of bone and muscle, excess fat, and aging can result in these unfavorable changes. Despite growing evidence of shared risk factors and mechanisms between osteoporosis, sarcopenia, and obesity [4] previous studies have only shown that lung function is associated with each of these components or with sarcopenic obesity. However, the present study demonstrated that individuals with any combination (one, two, or three abnormal body compositions, especially OSO) had a greater risk of poor lung function than those with no abnormal body composition. These findings suggest that OSO and the coexistence of these abnormal body compositions can be deteriorating factors leading to poor lung function.

In this study, there was a significant relationship between abnormal body composition only in the RSP group and no significant association with OSP. Similar to our results, Hickson et al. had shown that pericardial fat has a positive association with RSP but no association with OSP [37]. There have been several studies and explanations for this result. A previous study showed that despite a significant difference in FEV1 and FVC, FEV1/FVC (a marker of OSP) was not different between patients with sarcopenic obesity and normal subjects [38]. As per previous studies, FEV1 and FVC are independently associated with low skeletal muscle mass, but there is no association with FEV1/FVC [34,35] In a study of a large group of Korean adults, low muscle mass was highly associated with FEV1 and FVC but not with FEV1/FVC [35]. Subjects with low skeletal muscle mass may have a weak ability to deflate and inflate lung volume, and restricted lungs may show reduced FVC and FEV1 values. Therefore, FVC and FEV1 values may be decreased in patients with sarcopenia; however, FEV1/FVC, a marker of OSP, remained unchanged. Additionally, as previously mentioned, adipose tissue reduces lung volume and capacity by pressing the chest wall in obese patients, and VC is thought to be reduced by kyphosis or secondary vertebral fractures in patients with osteoporosis. Therefore, multiple abnormal body composition parameters have a greater association with RSP than with OSP.

The baseline characteristics such as age, sex, BMI, SBP, behavioral factors, comorbidities, cholesterol level, insulin level, and HbA1c level had significant trend according to the number of abnormal body compositions. It is known that there are metabolic and behavioral differences according to abnormal body compositions, especially in OSO group which has triple adverse body compositions [39]. For this reason, there would be such significant differences between our study groups. Therefore, we tried to adjust possible confounding factors, and even after the adjustment, there were significant differences in RSP participants.

This study has several limitations. First, its cross-sectional design precludes the determination of a causal relationship. Second, sarcopenia was defined as loss of muscle mass because muscle strength and physical performance were not assessed in the study design. Third, skeletal muscle mass was measured using BIA, which depends on the relationship between body composition and water content. It can be inaccurate in some conditions. However, recent studies have shown that skeletal muscle mass analyzed by BIA correlates well with that analyzed by DEXA, a well-validated technique for the analysis of body composition. We evaluated BIA only in the 8-h fasting state, and this was applied uniformly to all participants. Fourth, because our study was conducted on reasonably healthy people who visited a health checkup center, it does not represent the entire Korean population. Fifth, because COPD patients presented an OSP in spirometry, it would be possible that some participants with COPD may have been included in this study. In diagnosis of COPD, however, clinical symptoms or exposure history as well as spirometry is important [40]. In this study, participants were not considered to have COPD because they were healthy adults who have received a health screening check-up.

In conclusion, this study showed that multiple abnormal body compositions, especially osteosarcopenic obesity, were independently associated with poor lung function and RSP in a Korean population over 50 years of age. Our findings suggest that decreased lung function may be predicted by the coexistence of abnormal body composition, especially OSO. Further longitudinal studies are necessary to investigate whether multiple abnormal body compositions affect poor lung function.

## Figures and Tables

**Figure 1 nutrients-15-02933-f001:**
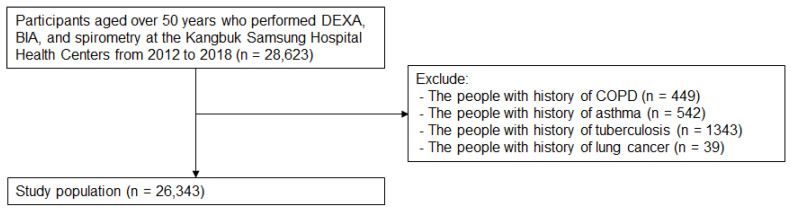
Selection of Study Population.

**Figure 2 nutrients-15-02933-f002:**
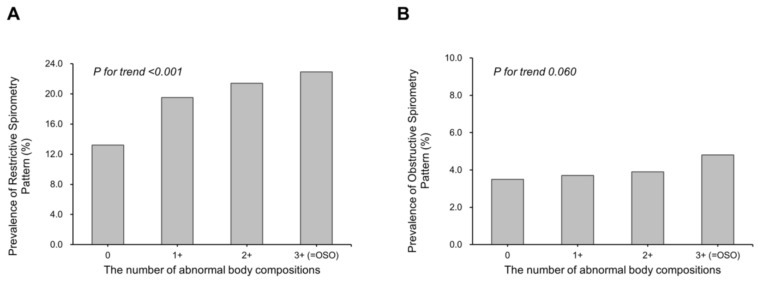
Prevalence of restrictive spirometry pattern (**A**) and obstructive spirometry pattern (**B**) according to abnormal body compositions. Restrictive spirometry pattern: FEV1/FVC ≥ 70% and FVC < 80% predicted. Obstructive spirometry pattern: FEV1/FVC < 70% and FEV1 < 80% predicted. OSO, osteosarcopenic obesity.

**Table 1 nutrients-15-02933-t001:** Participants characteristics by number of abnormal body compositions.

Characteristics	Total	No. of Abnormal Body Compositions ^a^				*p* for Trend
		0	1+	2+	3+ (=OSO)	
Number of subjects	26,343	8497	11,895	5290	661	
Demographic factors						
Age (year)	58.9 (6.7)	56.5 (5.6)	59.1 (6.5)	61.7 (7.0)	63.5 (8.0)	<0.001
Center (Seoul)	73.7	75.8	73.8	71.0	68.5	<0.001
Men (%)	32.8	41.2	33.8	19.4	16.7	<0.001
Height (cm)	160.5 (7.8)	162.6 (7.6)	160.4 (7.6)	157.8 (7.4)	157.2 (8.3)	<0.001
Body weight (kg)	61.0 (10.1)	60.9 (8.8)	62.2 (11.0)	59.3 (9.7)	56.9 (7.2)	<0.001
BMI (kg/m^2^)	23.6 (2.9)	22.9 (2.0)	24.0 (3.2)	23.7 (3.4)	22.9 (1.6)	<0.001
Waist circumference (cm)	81.9 (8.6)	79.7 (6.9)	83.0 (9.2)	82.9 (9.4)	82.6 (6.3)	<0.001
Fat mass (%)	29.9 (7.0)	26.5 (5.5)	30.6 (6.7)	32.9 (7.6)	34.4 (5.8)	<0.001
ASM (kg)	17.5 (4.0)	18.5 (4.0)	17.6 (3.9)	15.9 (3.3)	14.9 (3.6)	<0.001
SMI ^b^ (kg/m^2^)	6.7 (0.9)	6.9 (0.9)	6.7 (0.9)	6.3 (0.8)	5.9 (0.9)	<0.001
SBP (mmHg)	113.7 (14.0)	112.0 (13.4)	114.5 (14.0)	114.7 (14.4)	114.6 (13.9)	<0.001
DBP (mmHg)	72.1 (9.5)	72.0 (9.5)	72.6 (9.6)	71.4 (9.4)	70.7 (8.9)	<0.001
Health behavioral factors						
Current smoker (%)	9.9	11.9	9.9	7.4	5.4	<0.001
Heavy drinking ^c^ (%)	15.8	19.8	16.2	9.4	6.3	<0.001
HEPA ^d^ (%)	22.7	27.9	21.8	17.6	11.5	<0.001
Comorbidities						
Hypertension (%)	26.1	20.4	28.7	28.8	31.0	<0.001
Diabetes (%)	9.2	8.6	9.7	9.3	8.5	0.228
Dyslipidemia (%)	30.5	25.8	31.9	34.0	37.1	<0.001
Laboratory findings						
Total cholesterol (mg/dL)	199.7 (38.3)	198.7 (36.6)	199.3 (38.7)	201.8 (39.8)	203.2 (41.6)	<0.001
LDL (mg/dL)	131.0 (35.9)	129.4 (34.2)	131.0 (36.2)	132.9 (37.4)	135.1 (39.2)	<0.001
HDL (mg/dL)	60.4 (16.4)	61.5 (16.6)	59.6 (16.2)	60.5 (16.5)	58.4 (15.9)	<0.001
TG (mg/dL)	110.9 (66.4)	105.5 (66.3)	113.4 (68.7)	112.8 (60.0)	120.7 (68.3)	<0.001
Fasting glucose (mg/dL)	99.4 (17.4)	97.8 (15.6)	99.8 (18.1)	100.5 (18.0)	101.8 (19.6)	<0.001
Insulin (uIU/mL)	6.1 (5.3)	5.3 (4.2)	6.5 (4.1)	6.6 (8.7)	6.5 (3.2)	<0.001
HbA1c (%)	5.8 (0.6)	5.7 (0.5)	5.8 (0.6)	5.8 (0.6)	5.9 (0.8)	<0.001

Values are mean (standard deviation), or percentage. ^a^ Number of osteoporosis/osteopenia, sarcopenia, and obesity. ^b^ SMI = appendicular skeletal muscle mass/height^2^ (m^2^). ^c^ over 20 g per day. ^d^ HEPA was defined as IPAQ class II. SMI, skeletal muscle mass index; SBP, systolic blood pressure; DBP, diastolic blood pressure; HEPA, health enhancing physical activity; IPAQ, International Physical Activity Questionnaire; LDL, low density lipoprotein cholesterol; HDL, high density lipoprotein cholesterol; TG, triglyceride; HOMA-IR, insulin resistance assessed by homeostasis model assessment; AST, aspartate aminotransferase; ALT, alanine aminotransferase; CRP, C-reactive protein.

**Table 2 nutrients-15-02933-t002:** Pulmonary function values of study subjects by abnormal body compositions.

Variables	No. of Abnormal Body Compositions				*p* for Trend
	0	1+	2+	3+ (=OSO)	
FVC, L	3.47 (0.74)	3.20 (0.70)	2.87 (0.61)	2.69 (0.60)	<0.001
Mean predicted FVC%	91.34 (10.66)	88.89 (11.18)	88.37 (11.74)	87.43 (12.01)	<0.001
FEV1, L	2.73 (0.55)	2.52 (0.53)	2.27 (0.47)	2.11 (0.45)	<0.001
Mean predicted FEV1%	91.50 (11.53)	90.58 (12.02)	90.70 (12.85)	89.74 (13.20)	<0.001
FEV1/FVC, %	78.96 (6.05)	79.19 (6.14)	79.54 (6.38)	78.91 (6.49)	0.756

Values are mean (standard deviation). OSO, osteosarcopenic obesity; FVC, forced volume vital capacity; FEV1, force expiratory volume in 1 s.

**Table 3 nutrients-15-02933-t003:** Binary logistic regression analysis showing association of RSP and OSP with the number of abnormal body compositions.

No. of Abnormal Body Compositions	Crude	Adjusted OR *
Restrictive spirometry pattern		
0	1.00 (reference)	1.00 (reference)
1+	1.59 (1.47–1.72)	1.36 (1.23–1.50)
2+	1.79 (1.63–1.96)	1.47 (1.30–1.67)
3+ (=OSO)	1.96 (1.64–2.33)	1.64 (1.27–2.11)
*p* for trend	<0.001	<0.001
Obstructive spirometry disease		
0	1.00 (reference)	1.00 (reference)
1+	1.07 (0.92–1.24)	0.92 (0.75–1.13)
2+	1.12 (0.93–1.34)	1.01 (0.77–1.32)
3+ (=OSO)	1.38 (0.97–1.95)	0.58 (0.28–1.21)
*p* for trend	0.060	0.488

* Adjusted for age, sex, center, SBP, insulin, HDL-C, current smoker, heavy drinking, and HEPA. Restrictive spirometry pattern: FEV1/FVC ≥ 70% and FVC < 80% predicted. Obstructive spirometry pattern: FEV1/FVC < 70% and FEV1 < 80% predicted.

## Data Availability

Not applicable.

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
