# Peer review of "Association of Pulmonary Function with Osteosarcopenic Obesity in Older Adults Aged over 50 Years"

_nutrients, 2023, doi:10.3390/nu15132933_

Round 1

Reviewer 1 Report

In this study, the authors investigated the association of osteosarcopenic obesity (OSO) with indicators of lung function (FCV and FEV1) in a sample of Koreans aged 50+ years. Subjects (26,343 of them) were divided into four groups according to the presence of none to a maximum of three indicators of changes in body composition, osteopenia/osteoporosis, sarcopenia and obesity (OSO), after which sociodemographic, anthropometric, biochemical and health indicators were presented for each group and compared between groups, as well as pulmonary function values of FVC, FEV1 and FEV1/FVC. It was shown that the more OSO components are present, the greater the decrease in pulmonary function values ​​(FVC and FEV1) was, just as a greater number of OSO components is positively associated with the prevalence of a restrictive spirometry pattern, but not with an obstructive respiratory pattern. Multivariate analyzes showed the same trend for the restrictive spirometry pattern compared to the control group. The presence of multiple components of ODO could be a predisposing factor for deterioration of lung function in elderly Koreans.

The study is adequately presented: previous research is adequately cited, the subjects and the methodology used are well described, the biostatistical methods are appropriate, the results are clearly presented, and the conclusions are based on the results. Also, the main limitations of this research are highlighted.

I just have some minor comments/objections:

page 2, line 50 – please correct “no study studies”

page 3, line 90 – please cite the latest paper on the recommendations of the Asian Working Group for Sarcopenia

page 4, line 145 – there is an extra space in “(n = 26,343)”

Table 1 – there is an extra dot in the “2+” column showing the mean height

page 5, line 183 – the Figure 2 is missing

page 5, line 190 – please add “ … and 1.96 in 3+”

page 6, line 196 – “The study of adults ….”

page 6, line 210 – “ … had significantly lower FVC, ….”

page 6, lines 219-220 – please rephrase this sentence to “It is a well-known fact that with aging there are changes in lung function, such as ….”

page 7, lines 265-268 – please delete

Minor editing is required.

Author Response

[Comment#1]

page 2, line 50 – please correct “no study studies”

Response: Thank you for your thoughtful comment. As you suggested, we corrected this sentence.

[Comment#2]

page 3, line 90 – please cite the latest paper on the recommendations of the Asian Working Group for Sarcopenia

Response: Thank you for your thoughtful comment. As you mentioned, we tried to find the latest AWGS recommendation, but we couldn't find it. To the best of our knowledge, the 2019 update we cited in this manuscript seems to be the latest paper. We would appreciate that you could let me know if you know about it.

[Comment#3]

page 4, line 145 – there is an extra space in “(n = 26,343)”

Response: Thank you for your thoughtful comment. As you suggested, we corrected this sentence.

[Comment#4]

Table 1 – there is an extra dot in the “2+” column showing the mean height

Response: Thank you for your thoughtful comment. As you suggested, we corrected this word.

[Comment#5]

page 5, line 183 – the Figure 2 is missing

Response: Thank you for your thoughtful comment. As you suggested, we added this word.

[Comment#6]

page 5, line 190 – please add “ … and 1.96 in 3+”

Response: Thank you for your thoughtful comment. As you suggested, we corrected this sentence.

[Comment#7]

page 6, line 196 – “The study of adults ….”

Response: Thank you for your thoughtful comment. As you suggested, we corrected this sentence.

[Comment#8]

page 6, line 210 – “ … had significantly lower FVC, ….”

Response: Thank you for your thoughtful comment. As you suggested, we corrected this sentence.

[Comment#9]

page 6, lines 219-220 – please rephrase this sentence to “It is a well-known fact that with aging there are changes in lung function, such as ….”

Response: Thank you for your thoughtful comment. As you suggested, we corrected this sentence.

[Comment#10]

page 7, lines 265-268 – please delete

Response: Thank you for your thoughtful comment. As you suggested, we deleted this sentence.

Reviewer 2 Report

1. The patients with a history of COPD  were excluded.  But in your cases included, "Restrictive spirometry pattern (RSP) was defined as FVC less than 80% predicted and 116 normal or increased FEV1/FVC . Obstructive spirometry pattern (OSP) was defined as 117 FEV1/ FVC <0.7 and predicted FEV1 <80%". Maybe there are some overlapping in included and excluded cases. Please define it more clearly. 

2. Table1: How to explain the significance of trend in different groups, such as BMI, heavy smoker, insulin and ALT?

The writing is easy to read. Thank you

Author Response

[Comment#1]

The patients with a history of COPD were excluded. But in your cases included, "Restrictive spirometry pattern (RSP) was defined as FVC less than 80% predicted and 116 normal or increased FEV1/FVC . Obstructive spirometry pattern (OSP) was defined as 117 FEV1/ FVC <0.7 and predicted FEV1 <80%". Maybe there are some overlapping in included and excluded cases. Please define it more clearly.

Response: Thank you for your thoughtful comment. In diagnosis of COPD, clinical symptoms or exposure history as well as spirometry is important [1]. In this study, however, participants may not be considered to have COPD because they are healthy adults who have received a health screening check-up. Nevertheless, as you mentioned, it would be possible that some participants with COPD may have been included. We added this limitation in the discussion section as follows. Additionally, to clarify this, we will define the OSP more clearly in the method section as follows.

[Discussion section: 5th paragraph] Fifth, because COPD patients presented an OSP in spirometry, it would be possible that some participants with COPD may have been included in this study. In diagnosis of COPD, however, clinical symptoms or exposure history as well as spirometry is important [39]. In this study, participants were not considered to have COPD because they were healthy adults who have received a health screening check-up.

[Method section: 8th paragraph] Obstructive spirometry pattern (OSP) was defined as FEV1/ FVC <0.7 and predicted FEV1 <80%. à Obstructive spirometry pattern (OSP) was defined as participants with FEV1/ FVC <0.7 and predicted FEV1 <80%, and no clinical symptoms.

[Comment#2]

How to explain the significance of trend in different groups, such as BMI, heavy smoker, insulin and ALT?

Response: Thank you for your thoughtful comment. As you said, the baseline characteristics such as age, sex, BMI, SBP, behavioral factors, comorbidities, cholesterol level, insulin level, and HbA1c had significant trend according to the number of abnormal body compositions. It is known that there are the metabolic and behavioral differences according to abnormal body compositions, especially in OSO group which has triple adverse body compositions [2]. For this reason, there were significant differences between our study groups. Therefore, we tried to adjust possible confounding factors, and even after the adjustment, there were significant differences in RSP participants. We added this contents in the discussion section as follows.

[Discussion section: 4th paragraph] The baseline characteristics such as age, sex, BMI, SBP, behavioral factors, comorbidities, cholesterol level, insulin level, and HbA1c level had significant trend according to the number of abnormal body compositions. It is known that there are the metabolic and behavioral differences according to abnormal body compositions, especially in OSO group which has triple adverse body compositions [39]. For this reason, there would be such significant differences between our study groups. Therefore, we tried to adjust possible confounding factors, and even after the adjustment, there were significant differences in RSP participants.